# SIFusion: Lightweight infrared and visible image fusion based on semantic injection

**Song Qian, Liwei Yang, Yan Xue, Ping Li** *

Faculty of Information Engineering, Xinjiang Institute of Technology, Aksu, China

* 17865563003@163.com

## Abstract

The objective of image fusion is to integrate complementary features from source images to better cater to the needs of human and machine vision. However, existing image fusion algorithms predominantly focus on enhancing the visual appeal of the fused image for human perception, often neglecting their impact on subsequent high-level visual tasks, particularly the processing of semantic information. Moreover, these fusion methods that incorporate downstream tasks tend to be overly complex and computationally intensive, which is not conducive to practical applications. To address these issues, a lightweight infrared and visible light image fusion method known as SIFusion, which is based on semantic injection, is proposed in this paper. This method employs a semantic-aware branch to extract semantic feature information, and then integrates these features into the fused features through a Semantic Injection Module (SIM) to meet the semantic requirements of high-level visual tasks. Furthermore, to simplify the complexity of the fusion network, this method introduces an Edge Convolution Module (ECB) based on structural reparameterization technology to enhance the representational capacity of the encoder and decoder. Extensive experimental comparisons demonstrate that the proposed method performs excellently in terms of visual appeal and advanced semantics, providing satisfactory fusion results for subsequent high-level visual tasks even in challenging scenarios.

## Introduction

Different imaging modalities have their own characteristics, and they describe image scenes in different ways [1]. Taking visible light cameras and infrared cameras as examples, the former generates images by reflecting light from objects, which has the advantage of capturing rich texture details. However, it is also susceptible to adverse imaging conditions, such as scene brightness and fog. Relatively speaking, infrared cameras rely on thermal radiation information from objects to image, so it performs well in highlighting targets, but it also has its limitations, such as low image resolution and relatively less background detail information [2]. It is precisely because of the natural complementary characteristics of these two modalities of sensors that many researchers have begun to explore methods of fusing infrared and visible light images to generate richer information fusion images.

Fig 1 shows a challenging example. As shown in Fig 1(a), in nighttime scenes, visible light images are affected by ambient lighting, making it difficult to recognize pedestrians and

**Data Availability Statement:** All relevant data are within the manuscript and its Supporting information files.

**Funding:** This research was funded by the Xinjiang Uygur Autonomous Region Tianchi Talent

Innovation Leading Talent Project of China (NO.2023TCLJ02). The funder participated in research design, data collection and analysis, publication decisions, and manuscript preparation.

vehicles. However, infrared images can clearly capture these targets, thanks to the unique imaging principle of infrared sensors. Fig 1(b) shows an advanced image fusion method [3], but the test results of the mainstream YOLOv5s object detector [4] show that in complex scenes, the method still needs to be improved in terms of incorporating semantic information into the fused image. Therefore, as shown in Fig 1(d), an ideal image fusion method should not only pursue good visual effects, but also ensure the integrity of source image information and highlight significant targets. Only in this way can image fusion technology play a greater role in various practical applications such as target tracking [5], nighttime assisted driving [6], pedestrian re-recognition [7], object detection [8] and semantic segmentation [9].

In recent years, the fusion technology of infrared and visible light images has received widespread attention from researchers. Early image fusion methods mainly rely on technologies such as multi-scale transformation [10], subspace transformation [11], sparse representation [12], and saliency analysis [13] to enhance visual effects. With the rapid development of deep learning, researchers have begun to combine convolutional neural networks (CNN), autoencoders (AE), and generative adversarial networks (GAN) to further improve the visual performance of fusion results. For example, Li et al. [14] added dense connections to the AE-based framework, enhancing the network's ability to extract features and making it easier to train. Jian et al. [15] introduced attention mechanisms into the AE, enabling the network to focus more on salient targets and texture details in the image. Tang et al. [16] improved the network's ability to interact with intermodal features by adding a cross-modality difference perception fusion module (CMDAF) to the CNN. Ma et al. [17] proposed STD FusionNet, which uses a salient object mask to select important information from infrared and visible light images. Ma et al. [18] were the first to apply GANs to the image fusion task, transforming the fusion task into an adversarial game between the generator and discriminator, but this method may be insufficient in preserving texture details. Subsequently, Ma et al. [19] designed DDcGAN, which designs a dual discriminator based on the modality differences between infrared and visible light images, but may lead to artifacts in some results. Although the aforementioned methods, especially those based on deep learning, have achieved good fusion results, they often overlook how to facilitate subsequent advanced visual tasks.

To address this issue, Tang et al. [3] proposed the semantic-aware fusion framework of SeAFusion, which enhances the semantic information in the fused image by attaching a segmentation model behind the fusion network. In addition, Tang et al. [20] also proposed PSFusion, in which the fusion network and segmentation network share the same semantic feature extraction network, thus better achieving semantic information fusion. Liu et al. [21] and Sun et al. [22] also designed image fusion methods based on object detection, aiming to force the fusion network to retain more semantic information from the perspective of detection. However, these methods mainly promote the inclusion of more semantic information in the fused image through high-level visual task models, which may impose certain limitations on the visual performance of the fused image. In addition, complex designs may lead to a significant

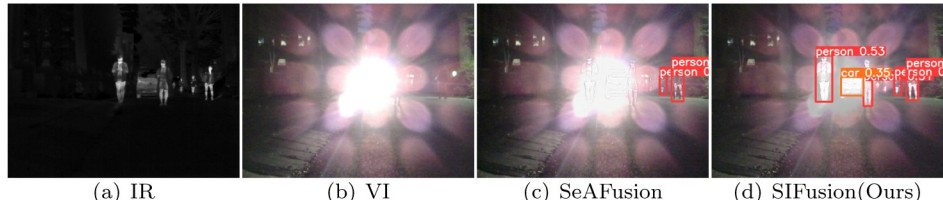

| (a) IR | (b) VI | (c) SeAFusion | (d) SIFusion(Ours) |

**Fig 1. Detection results of SeAFusion and SIFusion in challenging scenarios.**

increase in the computational load of network models, which is not conducive to the application of image fusion technology in practical engineering scenarios.

The design of lightweight models allows image fusion technology to better adapt to real-world scenarios, but such networks often lead to a decrease in fusion performance. To address this issue, researchers have proposed a series of innovative methods. IFCNN [23] uses only two convolutional layers for feature extraction and image reconstruction in its encoder and decoder. This method adjusts the fusion rules based on the type of source image, enabling a unified network to solve various fusion tasks. SDNet [24] generates fused images by constructing a squeeze-and-excitation network structure, ensuring that the fused image contains more source image information. SeAFusion [3] uses gradient residual dense blocks for feature extraction and combines semantic segmentation task loss to guide the training of the fusion network. Xue et al. [25] designed a fast and lightweight fusion network that simultaneously performs feature extraction and fusion. Chen et al. [26] designed a lightweight fusion network based on structural re-parameterization. Lu et al. [27] proposed LDRepFM, which achieves a balance between fusion speed and evaluation metrics through real-time end-to-end hierarchical decomposition and re-parameterization networks. Although structural re-parameterization technology can effectively solve the problem of computational resource consumption and imbalance in fusion performance, it still fails to fully focus on preserving semantic information. Therefore, to achieve effective preservation of semantic information, further research and optimization are needed for lightweight image fusion networks.

In view of the limitations of existing image fusion algorithms, this paper proposes a lightweight infrared and visible light image fusion network SIFusion based on semantic injection. The method includes a semantic feature extraction branch and an image information fusion branch, which effectively integrates modality image information and semantic information through a unique architecture. Then, this paper uses the semantic injection module (SIM) to fuse semantic features with heterogeneous modality image features, ensuring that the fused image has rich semantic clues. In addition, this paper introduces structural re-parameterization technology for optimizing the encoder and decoder, which not only improves the fusion performance, but also significantly reduces the number of required parameters and reduces the computational resource consumption. The main contributions of this paper are as follows:

- A lightweight infrared and visible image fusion framework based on semantic injection is proposed in this paper, which recognizes semantic features in multimodal images and effectively integrates them into the fusion result.

- A semantic injection module (SIM) is designed in this paper, which integrates semantic features with heteromodal image features, thus ensuring that the fused image is rich in semantic cues.

- The edge convolution block (ECB) based on structural reparameterization technique is introduced in this paper as an encoder and decoder, which significantly improves the fusion performance without increasing the computational burden in the inference phase.

- Numerous experiments demonstrate that the fusion results of the proposed method have good visual perception and advanced semantics, and outperform existing fusion algorithms in terms of fusion performance.

The remainder of this paper is organized as follows. In Section 2, this article briefly introduces related work on image fusion and semantic injection. In Section 3, this article details the proposed SIFusion, including the overall framework, semantic injection module, and loss function. In Section 4, through a large number of experimental comparisons, this method is

shown to have superior performance compared to other methods. Some conclusions are then drawn in Section 5.

## Related work

### Infrared and visible fusion

In early research, the main purpose of infrared and visible light image fusion was to ensure that the fusion results could fully present the information of the source images and make them more consistent with the human visual perception system. To better learn image features, people first considered training encoders and decoders using large-scale datasets. Li et al. [14] proposed a pretrained fusion model called DenseFuse, which used a natural image dataset to train the encoder and decoder, and then used fusion rules to combine the information of different modal images. Tang et al. [28] designed multiple encoders based on the Retinex theory to decompose the lighting and reflection components of visible light images, thereby enhancing the fusion effect in night scenes. To further achieve end-to-end image fusion, researchers also designed many unique loss functions and network architectures. Ma et al. [17] designed a fusion loss based on significant target masks, which can guide the fusion network to selectively process salient objects and background regions. Tang et al. [16] also constructed an illumination-aware subnetwork and a cross-modal differential perception fusion module to ensure that the fusion results are visually appealing. To address the challenge of simulating cross-modality features and decomposing ideal modality-specific and modality-shared features, Zhao et al. [29] proposed a novel Correlation-Driven feature Decomposition Fusion network, CDDFuse, which achieves good fusion effects through a dual-branch Transformer-CNN feature extractor. Due to the lack of real reference images, researchers also introduced generative adversarial mechanisms for unsupervised learning. Ma et al. [18] proposed FusionGAN, which was the first to introduce a generative adversarial network to the field of image fusion, aiming to further preserve more texture details and saliency. To alleviate the modal imbalance problem caused by a single discriminator, they also designed a dual discriminator-based DcGAN [19]. Wang et al. [30] proposed ICAFusion, which constructs a triple-path interactive compensatory attention fusion network to enhance the model's ability to extract global feature information. These early studies have laid a solid foundation for subsequent image fusion technologies and promoted the development of this field.

Although complex network designs can improve fusion performance to some extent, they also increase the difficulty of applying image fusion technology in real-world scenarios. To address this issue, some researchers have focused on developing lightweight fusion networks and carefully designed various network architectures. Tang et al. [3] proposed the SeAFusion method, which uses dense blocks of gradient residuals for feature extraction, aiming to improve the efficiency of feature extraction. Xue et al. [25] designed a fast and lightweight fusion network called FLFuse, which simultaneously performs feature extraction and feature fusion, thereby improving the fusion speed. Chen et al. [26] designed a lightweight fusion network based on structural re-parameterization technology, which significantly improves fusion performance without increasing the computational burden during the inference stage. Lu et al. [27] proposed LDRepFM, which combines real-time end-to-end hierarchical decomposition networks with re-parameterization networks to achieve a balance between fusion speed and evaluation metrics. Considering our need to improve fusion performance while maintaining lightweightness, this article decides to introduce structural re-parameterization technology to further enhance the feature extraction capability of the network. This technology can optimize network structure, reduce redundant parameters, and improve the efficiency and accuracy of feature extraction, thereby achieving better image fusion results.

## Semantic guidance

In recent years, some researchers have proposed practical solutions to enhance the deep learning networks' ability to extract semantic information by using semantic guidance maps. Wang et al. [31] achieved more realistic texture recovery effects by injecting semantic features into the super-resolution network through spatial feature transformation. Tang et al. [3] constructed a framework that integrates fusion branches with segmentation branches, guiding the training of the fusion network through segmentation loss, which allows the segmentation task to promote the performance improvement of the fusion task. The SCFusion proposed by Liu et al. [32] enhances the target prominence of the fusion results by fusing infrared salient information into the texture extraction branch network through spatial biasing. Liu et al. [33] proposed a multi-interactive feature learning architecture for image fusion and segmentation, SegMiF, and leveraged the correlation between dual tasks to enhance the performance of both.

However, these methods have not achieved effective semantic injection efficiently and lightweight. Therefore, this paper's approach draws on the network design of SCFusion and proposes a lightweight infrared and visible image fusion framework based on semantic injection. To more easily obtain semantic feature information, this framework utilizes semantic mask-guided semantic extraction encoder training to extract semantic features from the source images. Furthermore, through the semantic injection module, this method effectively integrates richer semantic information into the fused image, thereby achieving excellent performance for advanced visual tasks.

## Methods

### Network architecture

The overall architecture of SIFusion is shown in Fig 2, which mainly includes the image information fusion branch and the semantic feature extraction branch. The specific process is as follows: First, given a pair of registered infrared (IR) and visible light (VI) images, these two images are stitched together. Then, through the fusion encoder, the fused feature $\phi_f$ is extracted from the stitched image. At the same time, through the semantic extraction encoder, semantic feature $\phi_s$ is extracted from another path. Subsequently, the semantic feature $\phi_s$ undergoes processing by the semantic injection module (SIM), interacting with the fused feature $\phi_f$. This process aims to integrate more semantic information into the fused feature, thereby enhancing the semantic richness of the fused image. Finally, the processed features undergo information interaction through the fusion decoder to reconstruct the fused image (Fusion). Throughout this process, the method uses a carefully designed loss function to guide the training of the network, ensuring that the synthesized fused image performs well in both visual perception and advanced semantic tasks. This lightweight infrared and visible light image fusion framework SIFusion based on semantic injection not only improves the performance of image fusion but also provides efficient solutions for various practical application scenarios while maintaining lightweightness.

In the field of infrared and visible image fusion, both single-branch and dual-branch structures are among the mainstream network architectures in existing fusion methods. The dual-branch structure can extract features from two different modalities separately before fusing them, offering better interpretability. In contrast, the single-branch structure obtains shared feature information with similar information domains through the same feature extraction approach. To further achieve a lightweight network model and in conjunction with the characteristics of the semantic injection architecture, the method proposed in this paper selects a

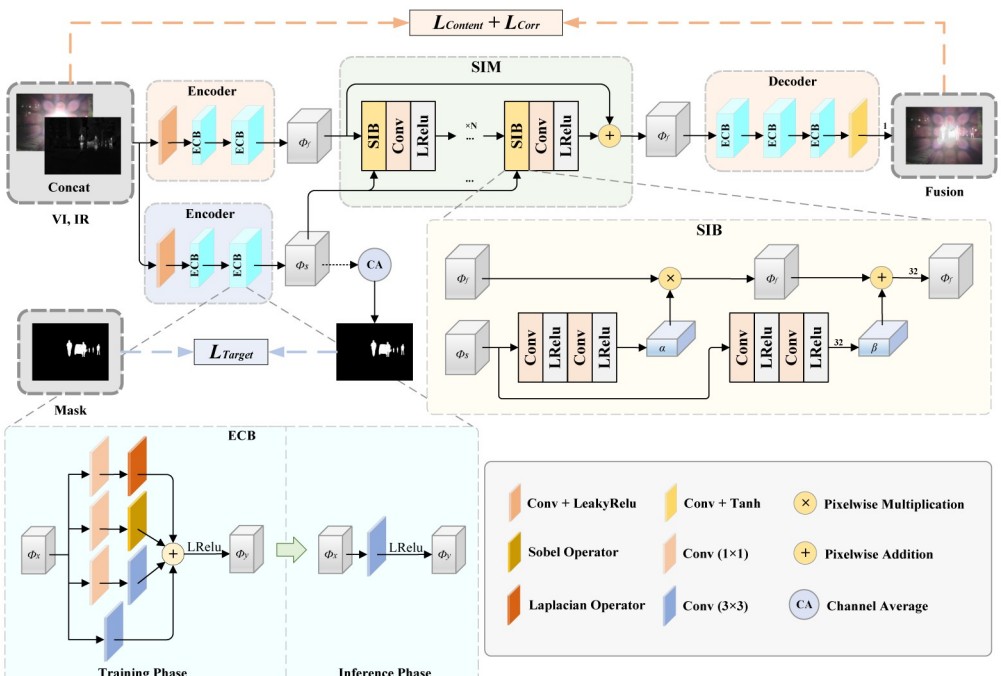

**Fig 2. The overall framework of SIFusion.** The figure include the Edge Convolution Module (ECB) and the Semantic Injection Module (SIM).

single-branch structure with fewer parameters, while the unique feature information is combined with the shared feature information through a semantic injection approach.

In addition, the main goal of this paper is to achieve lightweight image fusion. In order to enhance the representational ability of the network model while minimizing additional inference computation, this paper references the efficient and lightweight ECBSR [34] method. This paper applies the edge convolution block (ECB) based on structural re-parameterization technique to the encoder and decoder. This design enables the fusion network to have stronger feature extraction ability while maintaining relatively few parameters. It is worth noting that ECB adopts different network structures in the training stage and inference stage, which fully complies with the definition of structural re-parameterization technique [35]. This design not only improves the performance of the network, but also significantly reduces the computational cost, making it more suitable for practical application scenarios. The transformation process of the structural reparameterization of ECB is shown in Fig 3.

In order to allow the network to extract more meaningful information, such as edge feature information of the source image, predefined Sobel and Laplacian operators are added to multiple branches in ECB. The definitions of these two operators are as follows:

$$Sobel: \quad D_x = \begin{bmatrix} +1 & 0 & -1 \\ +2 & 0 & -2 \\ +1 & 0 & -1 \end{bmatrix} \quad and \quad D_y = \begin{bmatrix} +1 & +2 & +1 \\ 0 & 0 & 0 \\ -1 & -2 & -1 \end{bmatrix}, \tag{1}$$

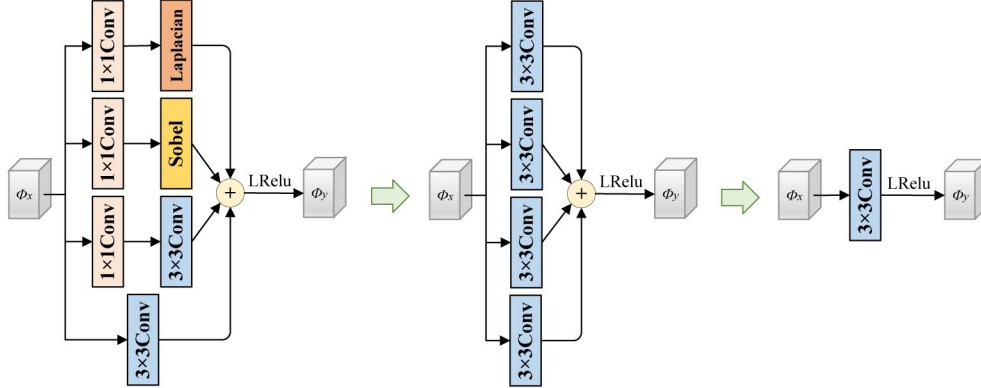

**Fig 3. The structural re-parameterization of ECB.** The figure include the transformation process of the structural re-parameterization of ECB.

$$Laplacian: \quad D_{lap} = \begin{bmatrix} 0 & +1 & 0 \\ +1 & -4 & +1 \\ 0 & +1 & 0 \end{bmatrix}, \tag{2}$$

where, two branches use Sobel and Laplacian operators, which can effectively extract edge feature information from images. Interestingly, these operators use the same computation method as DWConv, so it is possible to use structural re-parameterization techniques to combine the parameters of multiple branches into a single common convolution parameter, resulting in a more concise network structure during the inference stage.

During the training phase, having more branches is beneficial for the network model to possess a richer feature representation. Therefore, ECB can be formulated as:

$$\hat{\phi} = Conv_{3\times3}(\phi) + Conv_{1\times1}(\phi) + Conv_{Sobel}(\phi) + Conv_{Lap}(\phi), \tag{3}$$

Upon completion of the training of the fusion network, in order to further enhance the model's inference speed and reduce the computational consumption of the network, the trained network is optimized using structural reparameterization techniques. The optimized ECB can be formulated as:

$$\hat{\phi} = Conv_{3\times3}(\phi), \tag{4}$$

By referring to Fig 3, Eqs 3 and 4, it can be observed that the structural reparameterization technique can effectively reduce the number of branches in the network, thereby further decreasing computational consumption. This is beneficial for the model to better accomplish lightweight image fusion tasks.

## Semantic injection module (SIM)

Image fusion is a fundamental visual task, aiming to better serve high-level visual tasks such as semantic segmentation and object detection. Semantic information is particularly important in these high-level visual tasks. If we only pursue the visual effect of fused images and ignore the aggregation of semantic information, it may lead to poor fusion results. To address this

issue, this paper introduces a deep semantic information injection and modulation component in the intermediate stage of the encoder and decoder. This design enables the fusion network to aggregate more meaningful semantic information, thereby improving the visual effect of fused images while better serving high-level visual tasks.

Specifically, as shown in Fig 2, we expect to obtain semantic features of the source image through a semantic encoder $\phi_s$ can be better injected into the fusion branch. To achieve this goal, this article draws inspiration from the design of Wang et al. [31] and develops an efficient Semantic Injection Module (SIM). This module consists of $N$ units (N = 3), each containing a semantic injection block (SIB) and $3 \times 3$ Convolution and LeakyRelu activation function. Among them, the Semantic Injection Block (SIB) is the core part of the entire module, which enables effective interaction between semantic features and fused features. Through the design of this module, the model can integrate more semantic information into the fused image, thus improving the adaptability of the fusion results to high-level visual tasks. SIB can be formulated as:

$$\alpha = Conv_\alpha^2(\phi_s), \beta = Conv_\beta^2(\phi_s),$$
$$\hat{\phi}_f = \phi_f \otimes \alpha \oplus \beta, \tag{5}$$

where, $\otimes$ represents the pixel multiplication, and $\oplus$ represents the pixel addition.

In order to better integrate semantics into the image information fusion branch, the Semantic Injection Module (SIM) fuses image information features and semantic information features through multiple Semantic Injection Blocks (SIBs) connected to convolutional pipelines. Additionally, residual connections are used to enhance the expression of source image feature information. This process can be formalized as:

$$\hat{\phi}_f{}' = Conv^N(\hat{\phi}_f) + \phi_f, N = \{1, 2, 3\}, \tag{6}$$

## Loss function

SIFusion not only directly constrains the fusion result through the fusion loss, but also uses the target loss to constrain the encoder network of the semantic feature extraction branch. Next, this section will also detail the fusion loss and target loss.

Fusion loss mainly includes content loss and correlation loss. In recent years, content loss functions have been widely used in CNN-type fusion networks. The definition of content loss function is as follows:

$$L_{centent} = L_{int} + \mu L_{grad}. \tag{7}$$

However, due to the characteristic of maximum intensity loss, when there are extreme environments such as strong light and fog in the visible light image, the fused image may learn more information from the visible light image. As a result, meaningful infrared image information may be overlaid by higher intensity visible light image information, which is not the desired result. In order to prevent the loss of target-level infrared information in complex scenes, this paper adopts a strategy of using a target Mask to preserve infrared intensity information. In this way, this paper can ensure that meaningful infrared information is not obscured by visible light information during the fusion process, thereby preserving more

target-level details in the final fused image. The detailed definition of intensity loss is as follows:

$$L_{in} = \frac{1}{HW}\|I_f - Max(I_{ir}, I_{vi}) \otimes (1 - I_{mask})\|_1 + \frac{1}{HW}\|I_f - I_{ir} \otimes I_{mask}\|_1. \qquad (8)$$

where, $\|\cdot\|_1$ represents $l_1 - norm$, and $\otimes$ represents the pixel multiplication.

In order to ensure that the fusion result can capture the texture details in the source image, this paper not only focuses on the intensity information of the source image, but also pays special attention to the edge gradient information. This is because the edge gradient plays a crucial role in the visual effect and texture recovery of the image. The detail definition of gradient loss is as follows:

$$L_{grad} = \frac{1}{HW}\||\nabla I_f| - Max(|\nabla Bila(I_{ir})|, |\nabla I_{vi}|)\|_1. \qquad (9)$$

where, $\nabla$ represents the Sobel operator, and $|\cdot|$ indicates the absolute operation.

In addition, the correlation loss is introduced in this paper to strengthen the correlation between the fused image and the source image, and the correlation loss is defined as follows:

$$L_{grad} = \frac{1}{corr(I_f, I_{ir}) + corr(I_f, I_{vi})}. \qquad (10)$$

where, $corr(\cdot)$ represents the correlation function.

In order to better extract the semantic information of the source image, this paper uses the target loss to constrain the training of the semantic coder. The target loss is defined as follows:

$$L_{target} = \frac{1}{HW}\|I_f - CA(\phi_s \otimes I_{mask})\|_1. \qquad (11)$$

where, $CA(\cdot)$ denotes the channel average function.

Finally, in this paper, the training of SIFusion is jointly constrained by multiple loss functions to obtain fusion results with better results. The overall loss is defined as follows:

$$L_{total} = \alpha L_{content} + \beta L_{corr} + \lambda L_{target}. \qquad (12)$$

## Experimental validation

### Experimental configurations

**Benchmark dataset.**   In order to verify the effectiveness of the proposed method, comparative experiments were conducted on three public datasets: MSRS [16], M3FD [21], and LLVIP [36]. In order to expand the data samples used for training and better train the model, a common method is to use the reshape operation, but this will destroy the continuity between adjacent pixels, which is not conducive to the model's learning of pixel texture details. Therefore, in this paper, the original $480 \times 640$ images in the training set on the MSRS dataset were cut into 16 pieces, which are $120 \times 160$ small blocks, thus expanding the source dataset from 1083 to 17328 pairs, including visible light images, infrared images, and corresponding masks. In addition to conducting comparative experiments on the MSRS dataset, this paper also conducted generalization experiments on the TNO and LLVIP datasets to verify the performance of SIFusion.

**Implementation details.**    The method in this article is an end-to-end model. The network optimizer uses AdamW, $epoch = 100$, $initial\ learning\ rate = 5 \times 10^{-4}$, and loss function

parameter is $\mu = 2$, $\alpha = 25$, $\beta = 25$, $\lambda = 50$. The test sets used are the public data sets MSRS, TNO, and LLVIP, which fuse infrared and visible light images. 30, 42, and 50 pairs of images are selected for algorithm comparison experiments. The entire experiment was implemented on the PyTorch deep learning framework on NVIDIA 2080Ti 11GB. All comparison algorithms in the experiment were set up according to the original paper.

**Comparison algorithm.** In this article, we compare SIFusion with three AE-based methods (DenseFuse [14], RFN-Nest [37], and CSF [38]), five CNN-based methods (SDNet [24], FLFuse [25], U2Fusion [39], SeAFusion [3], and PSFusion [20]), and four GAN-based methods (FusionGAN [18], GANMcC [40], TarDAL [21], and UMF-CMGR [41]).

**Evaluation metrics.** Because the task of infrared and visible light image fusion does not have a reference image, a single evaluation metric is not sufficient to prove the superiority of the fusion effect. Therefore, six general image quality evaluation metrics are introduced in this article, namely SD, MI, VIF, SCD, EN, and Qabf, to measure the effect of the fusion result from different perspectives. MI, SCD, and EN evaluate the amount of information contained in the fused image from the perspective of information quantity. SD measures the high contrast of the fused image from the perspective of contrast. Qabf measures the edge intensity retained in the fused image. VIF quantifies the amount of shared information, thereby measuring the degree to which the fusion result conforms to human visual perception. These six evaluation metrics are all positive indicators, that is, higher values represent better results.

## Comparison experiments

Figs 4 and 5 show the visualization results of the proposed method and twelve comparative algorithms. The red box highlights the degree of preservation of salient objects by each method, while the green box shows the differences in background details between different methods.

The MSRS dataset contains infrared and visible light images of urban street scenes in both daytime and nighttime. As shown in the visualization results in Fig 4, in daytime scenes, SIFusion exhibits significant advantages in visual effects compared to other methods. FusionGAN has a problem of texture blurring, while SDNet has clearer texture, but the overall brightness is

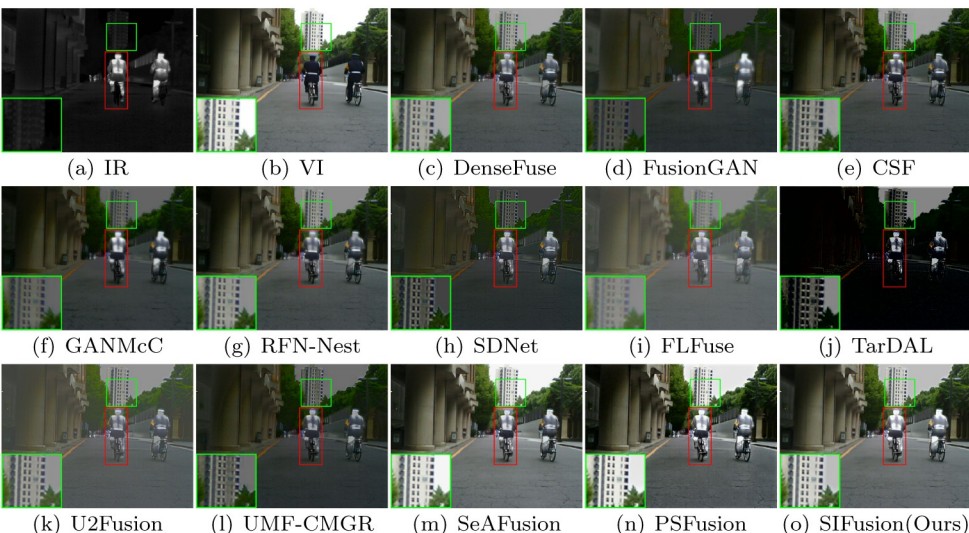

**Fig 4. Visualization results for daytime scenes in the MSRS dataset.**

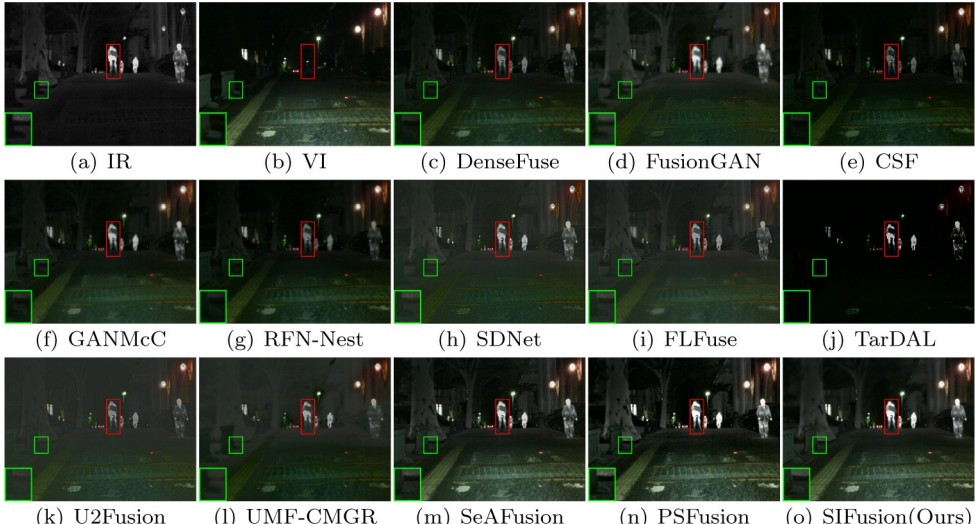

**Fig 5. Visualization results of the blackout scene in the MSRS dataset.**

insufficient. FLFuse and U2Fusion have insufficient contrast. The results of DenseFuse, CSF, and RFN-Nest achieve good visual effects to some extent, but the contrast and brightness of the scene are still insufficient. Although TarDAL can highlight the significance of semantic targets, the texture details of the background are distorted. In nighttime scenes, as shown in Fig 5, the above methods have similar limitations. Despite these algorithms being able to preserve the saliency of pedestrians in infrared images to varying degrees, SeAFusion, PSFusion, and SIFusion are all capable of making the background texture clearer and enhancing the visual experience while almost maintaining the saliency of semantic targets.

Visual qualitative comparison can distinguish the differences between different methods through human vision, but due to different display devices, the discrimination of some results may be relatively low, so quantitative comparison is needed in evaluation metrics.

The quantitative comparison results in the MSRS dataset are shown in Table 1. From the results, it can be seen that SeAFusion, PSFusion, and SIFusion belong to semantic-aware types of methods, which have better performance than other fusion methods. This indicates that the fused images generated by these methods contain rich information and transfer a substantial amount of information from the source images. Among them, PSFusion, which has the most complex model, has the highest evaluation metrics, while the evaluation metrics of SeAFusion and SIFusion are not significantly different, implying that SIFusion has a similar capability to retain semantic information as SeAFusion. However, compared to PSFusion and SeAFusion, SIFusion has a lighter network design, with specific model parameter comparisons shown in Table 6. In summary, the results of both qualitative and quantitative analysis can prove the superiority of SIFusion on the MSRS dataset.

## Generalization experiments

In addition to the experiments on the MSRS dataset, in order to validate the generalization of the methods in this paper, we also conducted experiments on the M3FD and LLVIP datasets.

**The M3FD dataset.** The M3FD dataset is a multi-scenario multi-modal dataset, featuring scenes in daylight, overcast, nighttime, and challenging environments, with rich semantic information of people, vehicles, etc. Figs 6 and 7 display the visualization results of the M3FD

**Table 1. Results of quantitative comparisons in the MSRS dataset.** The best results are marked in bold, the second-best results are underlined, and the third-best results are italicized.

| Algorithm | Evaluation Method | | | | | |
|---|---|---|---|---|---|---|
| | SD | MI | VIF | SCD | EN | Qabf |
| DenseFuse | 7.5090 | 2.6097 | 0.7317 | 1.4668 | 6.0225 | 0.4283 |
| FusionGAN | 5.7942 | 1.9529 | 0.4671 | 1.0191 | 5.4631 | 0.1524 |
| CSF | 6.9689 | 2.3774 | 0.6118 | 1.3640 | 5.6436 | 0.3221 |
| GANMcC | 8.0840 | 2.5387 | 0.6283 | 1.4622 | 6.0204 | 0.3013 |
| RFN-Nest | 7.5429 | 2.4150 | 0.6662 | 1.4611 | 5.9474 | 0.2927 |
| SDNet | 5.6207 | 1.8143 | 0.4149 | 1.0382 | 5.1713 | 0.3633 |
| FLFuse | 6.9163 | 2.2100 | 0.6698 | 1.2816 | 5.7258 | 0.3554 |
| TarDAL | 4.4567 | 1.1859 | 0.2315 | 0.8114 | 3.4366 | 0.1337 |
| U2Fusion | 5.7280 | 1.9882 | 0.3902 | 0.9897 | 4.7535 | 0.2845 |
| UMF-CMGR | 5.9766 | 1.9698 | 0.3836 | 0.9741 | 5.5499 | 0.2764 |
| SeAFusion | *8.1205* | **3.8122** | <u>0.9570</u> | *1.7795* | <u>6.5281</u> | <u>0.6298</u> |
| PSFusion | **8.2107** | *2.9234* | **1.0638** | **1.8315** | **6.7084** | **0.6467** |
| SIFusion (Ours) | <u>8.1275</u> | <u>3.0734</u> | *0.8373* | <u>1.8254</u> | *6.5104* | *0.5419* |

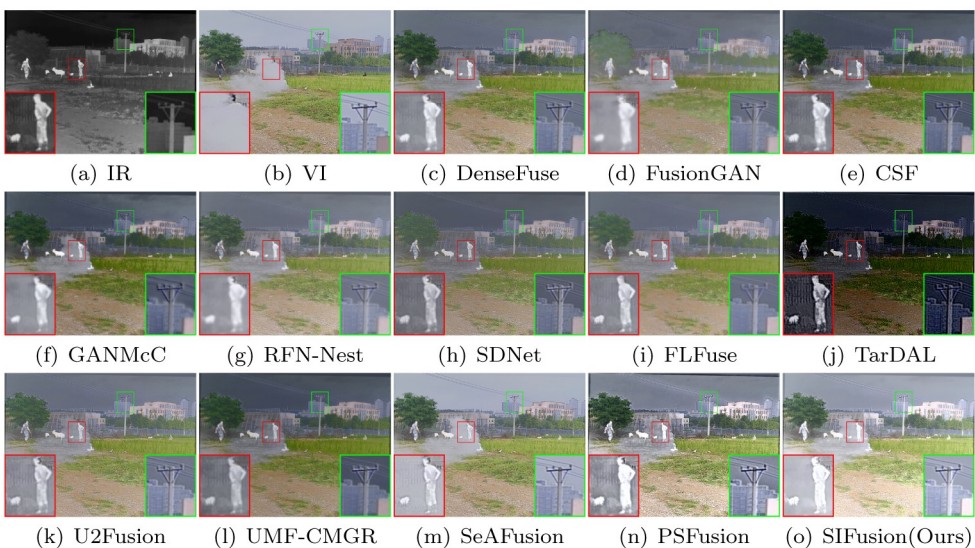

(a) IR          (b) VI          (c) DenseFuse          (d) FusionGAN          (e) CSF

(f) GANMcC          (g) RFN-Nest          (h) SDNet          (i) FLFuse          (j) TarDAL

(k) U2Fusion          (l) UMF-CMGR          (m) SeAFusion          (n) PSFusion          (o) SIFusion(Ours)

**Fig 6. Visualization results on the M3FD dataset.**

dataset, where the red-boxed areas are salient targets, and the green-boxed areas are background details. In terms of the saliency degree of pedestrians, the targets of TarDAL, PSFusion, and SIFusion are all quite prominent. However, the fusion results of TarDAL are too close to the infrared images, leading to the loss of some background texture information in the visible light images. In contrast, PSFusion and SIFusion can better preserve the scene description in the visible light images, especially in terms of background texture details.

In Table 2, PSFusion has the best performance, with SIFusion and SeAFusion ranking second and third in overall performance, respectively. We can see that SIFusion achieved good results in the three metrics of SD, SCD, and EN, following PSFusion, indicating that the proposed method can generate fusion results with better saliency. At the same time, it also

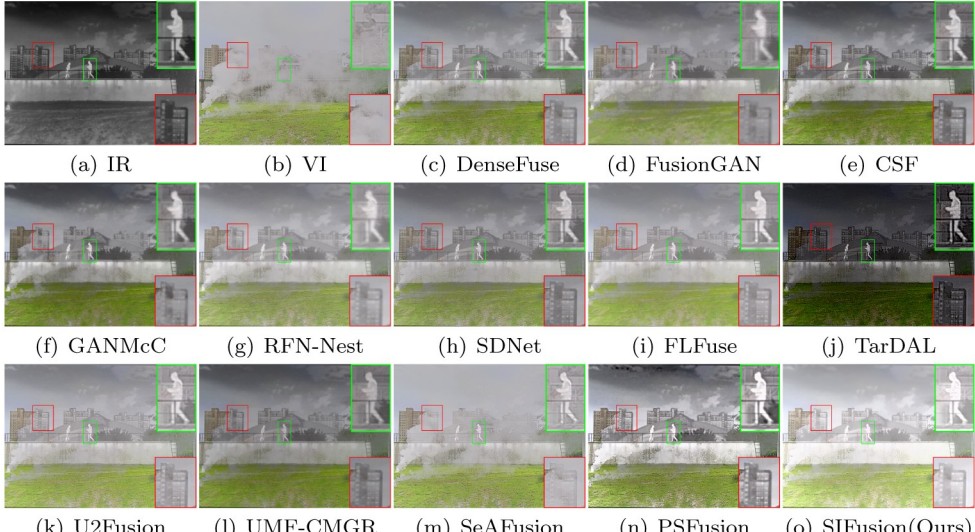

**Fig 7. Visualization results on the M3FD dataset.**

**Table 2. Results of quantitative comparisons in the M3FD dataset.** The best results are marked in bold, the second-best results are underlined, and the third-best results are italicized.

| Algorithm | Evaluation Method | | | | | |
|---|---|---|---|---|---|---|
| | SD | MI | VIF | SCD | EN | Qabf |
| DenseFuse | 9.5467 | 2.9715 | 0.8808 | 1.5240 | 6.9198 | 0.5083 |
| FusionGAN | 8.8368 | 2.8651 | 0.5800 | 0.7305 | 6.4545 | 0.2757 |
| CSF | 9.6359 | 2.9416 | 0.8938 | *1.5654* | *7.0076* | 0.4733 |
| GANMcC | 9.7510 | 2.8862 | 0.8084 | 1.3957 | 6.8776 | 0.3628 |
| RFN-Nest | 9.6196 | 2.9066 | 0.8466 | 1.4642 | 6.9160 | 0.3404 |
| SDNet | 9.3117 | *3.2765* | 0.7517 | 1.0709 | 6.7397 | 0.5119 |
| FLFuse | 9.4282 | **3.3695** | 0.8087 | 1.2593 | 6.8111 | 0.3028 |
| TarDAL | 8.9425 | 2.0294 | 0.7517 | 1.1622 | 6.4893 | 0.2725 |
| U2Fusion | 9.4868 | 2.8761 | 0.8224 | 1.3244 | 6.7633 | 0.5372 |
| UMF-CMGR | 9.5780 | 3.1377 | 0.8071 | 1.3363 | 6.8253 | 0.4048 |
| SeAFusion | *9.9133* | <u>3.3338</u> | <u>0.9766</u> | 1.4657 | 6.8788 | **0.5860** |
| PSFusion | **10.1806** | 2.8808 | **1.1700** | **1.8349** | **7.5824** | *0.5638* |
| SIFusion (Ours) | <u>10.1671</u> | 2.9759 | *0.9287* | <u>1.8013</u> | 7.2682 | <u>0.5781</u> |

performed well in the VIF and Qabf metrics, indicating that its fusion results are more visually compatible with the human visual system. In summary, through experimental verification with the M3FD dataset, SIFusion has shown good advantages in the fusion of infrared and visible light images in multi-scenarios.

**The LLVIP dataset.** LLVIP is a public dataset of infrared and visible light images for urban transportation in nighttime scenes. The images in this dataset have high image quality and contain a large number of common semantic objects, such as pedestrians and vehicles. Visualization results on the LLVIP dataset are shown in Figs 8 and 9.

The LLVIP is a public dataset of infrared and visible light images for urban traffic at night, characterized by high-quality images and containing a multitude of common semantic targets,

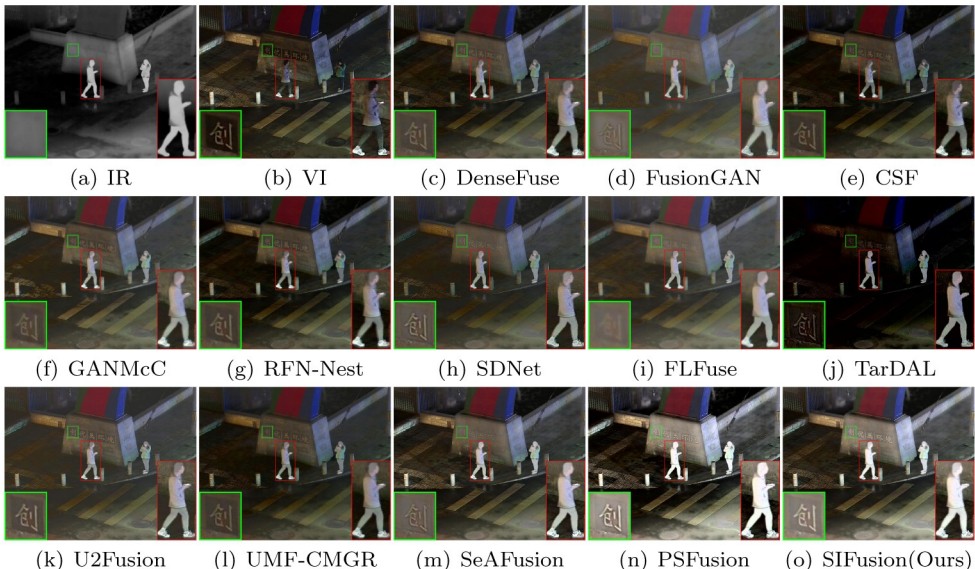

**Fig 8. Visualization results on the LLVIP dataset.**

such as pedestrians and vehicles. The visualization results on the LLVIP dataset are shown in Figs 8 and 9. From the magnified red boxes, it can be observed that, with the exception of SIFusion and PSFusion, other methods have to varying degrees weakened the contrast of the infrared targets. Meanwhile, from the magnified green boxes, it can be seen that DenseFuse, GANMcC, FLFuse, TarDAL, and SeAFusion have blurred the text in the background, while other methods, although capable of displaying details in the background, do not match the clarity of SIFusion and PSFusion.

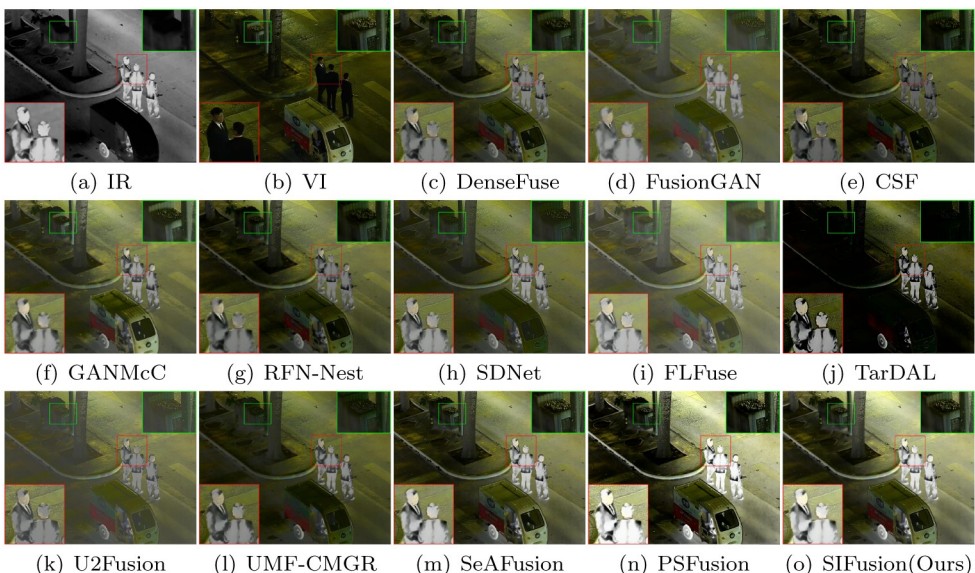

**Fig 9. Visualization results on the LLVIP dataset.**

**Table 3. Results of quantitative comparisons in the LLVIP dataset.** The best results are marked in bold, the second-best results are underlined, and the third-best results are italicized.

| Algorithm | Evaluation Method | | | | | |
|---|---|---|---|---|---|---|
| | SD | MI | VIF | SCD | EN | Qabf |
| DenseFuse | 9.2490 | 2.6993 | 0.8317 | 1.4190 | 6.8727 | 0.4774 |
| FusionGAN | 8.3299 | 2.8222 | 0.5322 | 0.7299 | 6.3078 | 0.2488 |
| CSF | 9.0546 | 2.5069 | 0.7944 | 1.3641 | 6.6971 | 0.4092 |
| GANMcC | 9.0244 | 2.6850 | 0.7155 | 1.2786 | 6.6894 | 0.2914 |
| RFN-Nest | 9.2655 | 2.5545 | 0.8196 | 1.4398 | 6.8622 | 0.3065 |
| SDNet | 8.9238 | 2.9773 | 0.6537 | 0.9803 | 6.6793 | 0.5056 |
| FLFuse | 9.2603 | 2.9975 | 0.7522 | 1.1367 | 6.7612 | 0.2744 |
| TarDAL | 7.4436 | 1.7770 | 0.4566 | 0.8075 | 5.1034 | 0.2340 |
| U2Fusion | 7.7951 | 2.4054 | 0.5631 | 0.8092 | 5.9464 | 0.3240 |
| UMF-CMGR | 8.0539 | 2.6817 | 0.5796 | 1.0481 | 6.4619 | 0.3416 |
| SeAFusion | *9.4885* | **3.7724** | <u>0.9882</u> | *1.5779* | *7.2353* | <u>0.6089</u> |
| PSFusion | <u>9.9358</u> | *3.0095* | **1.1044** | <u>1.6784</u> | **7.6017** | *0.5680* |
| SIFusion (Ours) | **10.0069** | <u>3.0883</u> | *0.9774* | **1.8194** | <u>7.5095</u> | **0.6141** |

Furthermore, the three metrics of SD, SCD, and Qabf in Table 3 are higher than other methods, indicating that the fusion results of SIFusion exhibit good contrast performance. Although the results in other metrics are not as good as PSFusion and SeAFusion, the proposed method achieves satisfactory performance with a lighter model. These results demonstrate that SIFusion has a distinct advantage in the fusion of infrared and visible light images for night urban traffic scenarios.

## Ablation experiment

In order to further validate the effectiveness of each module designed in the methodology of this paper, ablation experiments were also conducted in this paper. The qualitative and quantitative results of the ablation experiments are shown in Fig 10 and Table 4, respectively.

**Semantic injection module (SIM).** The semantic injection module (SIM) utilizes semantic information injection and modulation to enable the fusion network to integrate more semantic information. As shown in Fig 10(b), the proposed method can still maintain high

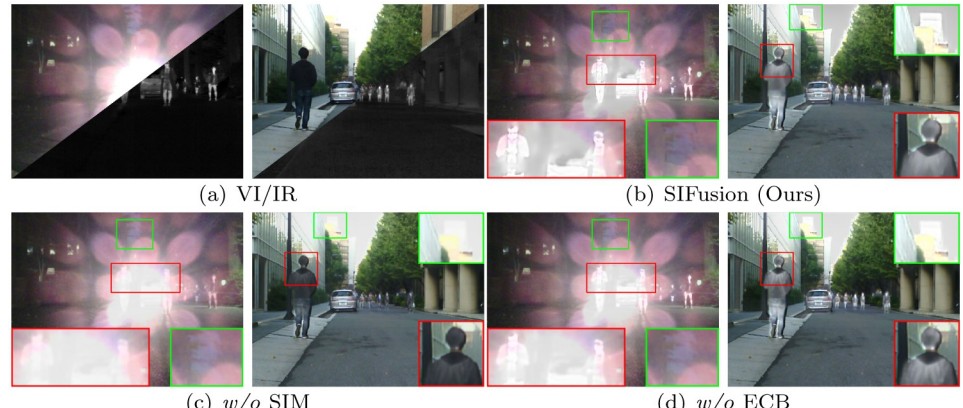

Fig 10. Ablation experiment visualization results.

**Table 4. Quantitative comparison results of ablation experiments.** The best results are marked in bold.

| Experiment | SD | MI | VIF | SCD | EN | Qabf |
|---|---|---|---|---|---|---|
| SIFusion (Ours) | **8.1275** | **3.0734** | **0.8373** | **1.8254** | **6.5104** | **0.5419** |
| -*w/o* SIM | 8.1033 | 3.0167 | 0.8073 | 1.7766 | 6.3528 | 0.4750 |
| -*w/o* ECB | 8.0992 | 3.0604 | 0.8347 | 1.8229 | 6.4795 | 0.5365 |

contrast of infrared targets in extreme scenarios. As shown in Fig 10(c), when the SIM is removed, the target contrast in the enlarged red box significantly decreases. At the same time, the values of the evaluation metrics in Table 4 also decrease. These results indicate that the semantic injection module has a promoting effect on the contrast preservation of infrared targets.

**Edge convolution block (ECB).** The Sobel operator and Laplacian operator are added to the Edge Convolution Block (ECB) to enhance the network's fine-grained expression. As shown in Fig 10(d), after removing the ECB, the enlarged background details in the green box become blurred, the gradient changes become less, and the values of the evaluation metrics in Table 4 decrease, further indicating that the performance is deteriorating. Both qualitative and quantitative results show the effect of this module on the overall network.

Additionally, to further validate the effectiveness of ECB, this paper also compares ECB with other typical modules, including 3 × 3 convolution, MobileNet Block [42], and GhostNet Block [43]. The comparison results are shown in Table 5. As indicated in Table 5, the Mobile-Net Block and GhostNet Block, as two typical modules, can achieve lightweight fusion to a certain extent. However, the ECB demonstrates better results across various evaluation metrics, which validates its effectiveness.

## Efficiency comparison experiment

In the methodology of this paper, attention is not only given to the quality of the fusion results but also to the lightweight nature of the model. To provide a more comprehensive assessment of the performance of SIFusion, this section takes reference from the image sizes in the MSRS dataset, setting the dimensions of the input data to 640 × 480 × 1. The relevant parameters and FLOPs of the network model are calculated using TorchSummary and Thop. The paper conducts an experimental comparison of SIFusion with twelve comparative algorithms in terms of runtime, operational memory space, parameter count, weight size, and FLOPs. The comparative results are shown in Table 6.

The data in Table 6 clearly shows that, with the exception of FLFuse, SIFusion has a significantly faster runtime compared to other comparative algorithms. Although FLFuse achieves the best performance in terms of speed, its lightweight model to some extent limits the extraction capability of source image information, leading to weaker performance in information integration. Additionally, the models of SeAFusion and PSFusion require a larger amount of parameters and computational resource consumption. In contrast, the method proposed in

**Table 5. Comparative experiment of ECB with other typical modules.** The best results are marked in bold.

| Experiment | SD | MI | VIF | SCD | EN | Qabf |
|---|---|---|---|---|---|---|
| ECB | **8.1275** | **3.0734** | **0.8373** | **1.8254** | **6.5104** | **0.5419** |
| 3 × 3 Conv | 8.0992 | 3.0604 | 0.8347 | 1.8229 | 6.4795 | 0.5365 |
| MobileNet Block | 7.8156 | 3.0467 | 0.8156 | 1.6397 | 6.2501 | 0.5392 |
| GhostNet Block | 7.7112 | 3.0681 | 0.8266 | 1.5836 | 6.2725 | 0.5335 |

**Table 6. The model parameters of SIFusion and comparative algorithms.** Bold indicates the best result and underline represents the second best result.

| Algorithm | Running time | Forward pass size | Params | Params size | FLOPs |
|---|---|---|---|---|---|
| DenseFuse | 0.2305s | <u>1129.69MB</u> | 88,225 | 0.34MB | 27.03G |
| FusionGAN | 0.1331s | 3379.69MB | 925,633 | 3.53MB | 284.50G |
| CSF | 2.1891s | 2885.16MB | 347,805 | 1.33MB | 106.86G |
| GANMcC | 0.1436s | 6754.69MB | 1864,129 | 7.11MB | 572.96G |
| RFN-Nest | 0.5273s | 15701.58MB | 7746,059 | 29.55MB | 520.85G |
| SDNet | 0.0198s | 1851.56MB | 93,971 | 0.36MB | 28.75G |
| FLFuse | **0.0015s** | **508.59MB** | **14,328** | **0.05MB** | **4.36G** |
| TarDAL | 0.0053s | 2179.69MB | 296,577 | 1.13MB | 91.14G |
| U2Fusion | 0.0928s | 2292.19MB | 659,217 | 2.51MB | 202.36G |
| UMF-CMGR | 0.0596s | 5414.06MB | 629,253 | 2.40MB | 193.21G |
| SeAFusion | 0.0178s | 4804.69MB | 166,657 | 0.64MB | 50.99G |
| PSFusion | 0.2294s | 13658.35MB | 45,899,360 | 175.09MB | 180.86G |
| SIFusion (Ours)* | 0.0382s | 4689.84MB | 84,115 | 0.33MB | 104.05G |
| SIFusion (Ours) | <u>0.0036s</u> | 2946.09MB | <u>83,557</u> | <u>0.32MB</u> | <u>25.52G</u> |

1. Ours* is SIFusion without re-parameterisation.

2. The difference in results between Ours* and Ours is $2 \times 10^{-3}$.

3. All model data are reproduced within the PyTorch architecture.

this paper, while maintaining a lower computational complexity, can effectively integrate information from the source images, thereby generating fused images that are more in line with the visual effects of human vision. This advantage endows SIFusion with higher efficiency and practicality in practical applications, especially in scenarios with limitations on real-time performance and computational resources. Therefore, the method in this paper excels in balancing network performance and lightweight design, providing a beneficial reference for the development of the lightweight image fusion field.

The method proposed in this paper significantly reduces the complexity and computational consumption of the SIFusion inference network through structural reparameterization techniques. This advantage makes SIFusion more practical and efficient in actual application scenarios. Therefore, in practical application scenarios, we can first train the fusion network on a high-computing platform and then optimize the network structure and computational consumption using structural reparameterization techniques. This process enables the deployment of the network on mobile devices with lower computational costs.

## Application of semantic segmentation

Although the method in this paper has good performance in the image quality and visibility of the fusion results, the fusion results still need to meet the semantic requirements of high-level machine vision. In order to verify the ability of the proposed method in semantic expression, more detailed experiments are conducted in this section. Specifically, twelve comparative algorithms are selected to compare with the proposed method in semantic segmentation task. To ensure fairness, the segmentation network is retrained using the MSRS dataset, and the configuration of the training set and test set is ensured to be consistent with that of SeAFusion [3]. Firstly, various fusion methods are used to generate fused images. Then, the pixel intersection-union (IoU) commonly used in semantic segmentation task is adopted as an evaluation metric to evaluate the segmentation performance of different fusion results. From Table 7, it can be seen that compared to the other twelve methods, the method proposed in this paper performs

**Table 7. Comparative experimental results for semantic segmentation performance.** Bold indicates the best result and underline represents the second best result.

| Label Class | Back ground | Car | Person | Bike | Curve | Car Stop | Guardrail | Color Cone | Bump | Average |
|---|---|---|---|---|---|---|---|---|---|---|
| VI | 0.9588 | 0.6825 | 0.2143 | 0.5525 | 0.2258 | 0.1953 | 0.4738 | 0.3119 | 0.1654 | 0.4268 |
| IR | 0.9422 | 0.2559 | 0.6587 | 0.0286 | 0.0735 | 0.0335 | 0.0142 | 0.0067 | 0.0083 | 0.2230 |
| DenseFuse | 0.9664 | 0.7305 | 0.6568 | 0.4967 | 0.2253 | 0.2032 | 0.4512 | 0.3173 | 0.1337 | 0.4646 |
| FusionGAN | 0.9603 | 0.6537 | 0.6956 | 0.3741 | 0.0770 | 0.1294 | 0.1894 | 0.2302 | 0.0330 | 0.3714 |
| CSF | 0.9662 | 0.7271 | 0.6455 | 0.4903 | 0.2222 | 0.2025 | 0.4318 | **0.3179** | 0.1251 | 0.4587 |
| GANMcC | 0.9646 | 0.7147 | 0.6789 | 0.4517 | 0.1580 | 0.1796 | 0.3495 | 0.3123 | 0.0873 | 0.4330 |
| RFN-Nest | 0.9672 | **0.7412** | 0.6539 | 0.5091 | 0.2381 | 0.2109 | 0.4265 | 0.3180 | 0.1350 | 0.4667 |
| SDNet | 0.9622 | 0.6736 | 0.6777 | 0.4363 | 0.0801 | 0.1524 | 0.3104 | 0.2805 | 0.0782 | 0.4057 |
| FLFuse | 0.9541 | 0.5926 | 0.6662 | 0.1953 | 0.1022 | 0.1155 | 0.2668 | 0.2322 | 0.0162 | 0.3490 |
| TarDAL | 0.9402 | 0.4400 | 0.3967 | 0.3166 | 0.0034 | 0.0798 | 0.2252 | 0.1717 | 0.0155 | 0.2877 |
| U2Fusion | 0.9589 | 0.6415 | 0.6171 | 0.4460 | 0.0909 | 0.1452 | 0.3658 | 0.2869 | 0.0938 | 0.4051 |
| UMF-CMGR | 0.9611 | 0.6729 | 0.6103 | 0.4078 | 0.1052 | 0.1480 | 0.2760 | 0.2923 | 0.0400 | 0.3904 |
| SeAFusion | 0.9652 | 0.7080 | 0.6663 | 0.5131 | <u>0.2416</u> | 0.1970 | <u>0.4939</u> | <u>0.3176</u> | 0.1537 | 0.4729 |
| PSFusion | **0.9681** | <u>0.7402</u> | **0.7209** | **0.5607** | **0.2665** | **0.2174** | **0.5082** | 0.3161 | **0.1701** | **0.4965** |
| SIFusion (Ours) | <u>0.9678</u> | 0.7362 | <u>0.7127</u> | <u>0.5557</u> | 0.2393 | <u>0.2134</u> | 0.4795 | 0.3163 | <u>0.1663</u> | <u>0.4874</u> |

quite well in segmentation accuracy across all categories, second only to PSFusion. This further proves the advantage of the method in this paper in enhancing the segmentation model's recognition of semantic information.

We believe that this excellent result is mainly due to two points: first, SIFusion can effectively integrate complementary information in infrared and visible light images, thereby helping the segmentation model to comprehensively understand the imaging scene. Secondly, the introduction of semantic injection module significantly enhances the expression of meaningful semantic information, making the fused image contain rich semantic information. In summary, improving the semantic information in the fused image is the key factor that makes our method superior to other fusion algorithms in terms of segmentation performance.

## Performance discussion on the TNO dataset

The TNO dataset is a classic dataset in the field of image fusion, involving a large number of military-related targets and scenes. Due to its early release time, the image data quality is poor, lacking effective background texture details and salient target information. To more comprehensively explore the performance of the methods presented in this paper, generalization comparative experiments were also conducted on the TNO dataset, and the results are shown in Table 8.

As can be seen from the results in Table 8, the fusion results of the method presented in this paper did not achieve satisfactory performance. This is due to the lack of common semantic targets such as pedestrians and vehicles in the TNO dataset, which led to the inability of our method to effectively extract the necessary semantic information on this dataset, resulting in poor fusion performance. Although our method has high requirements for the quality of the input images, as a lightweight image fusion algorithm, it still demonstrates good performance and great potential.

## Conclusion

In this study, we propose a lightweight semantic-infused fusion network framework called SIFusion. The framework can adaptively integrate meaningful semantic information by

**Table 8. Results of quantitative comparisons in the TNO dataset.** The best results are marked in bold, the second-best results are underlined, and the third-best results are italicized.

| Algorithm | Evaluation Method | | | | | |
|---|---|---|---|---|---|---|
| | SD | MI | VIF | SCD | EN | Qabf |
| DenseFuse | 9.2424 | 2.3019 | 0.8175 | 1.7838 | 6.8193 | 0.4457 |
| FusionGAN | 8.6736 | 2.3352 | 0.6541 | 1.3793 | 6.5580 | 0.2341 |
| CSF | 8.9505 | 2.0683 | 0.7946 | 1.7837 | 6.9184 | 0.3960 |
| GANMcC | 9.0532 | 2.2732 | 0.7123 | 1.7030 | 6.7359 | 0.2802 |
| RFN-Nest | 9.3589 | 2.1184 | 0.8183 | 1.7843 | 6.9632 | 0.3342 |
| SDNet | 9.0698 | 2.2606 | 0.7592 | 1.5590 | 6.6948 | 0.4294 |
| FLFuse | 9.2628 | 2.1925 | 0.8084 | 1.7338 | 6.3658 | 0.4177 |
| TarDAL | 9.3637 | 1.5212 | 0.7353 | 1.3432 | 6.5447 | 0.2036 |
| U2Fusion | 8.8553 | 1.8730 | 0.6787 | 1.5862 | 6.4230 | 0.4245 |
| UMF-CMGR | 8.7085 | 2.2140 | 0.7121 | 1.6354 | 6.5325 | 0.4099 |
| SeAFusion | **9.5693** | **2.8382** | **0.9811** | 1.7281 | 7.1335 | 0.4872 |
| PSFusion | 9.4278 | 2.3082 | 0.9321 | **1.8157** | **7.2529** | **0.5223** |
| SIFusion (Ours) | 9.5148 | 2.0840 | 0.7394 | 1.6988 | 6.7181 | 0.4537 |

designing a semantic injection module (SIM) to inject semantic feature information into fused features, and introducing an edge convolution block (ECB) based on structural re-parameterization technique to achieve high-performance lightweight image fusion. At the same time, we also design content loss, similarity loss, and target semantic loss based on the mask of salient objects to better achieve the desired results. A large number of experiments have shown that SIFusion can handle various complex scenarios well.

However, in low-light scenarios, due to the severe degradation of visible light images, almost all fusion methods, including the method in this paper, have the limitation of not being able to effectively extract feature information from visible light images. A potential solution is to combine SIFusion with low-light enhancement techniques to achieve semantic-driven fusion in low-light scenarios. In addition, in the future, we can further improve SIFusion to meet the real-time demands of complex scenes in video image fusion, which has significant application value in the field of security surveillance.

## Supporting information

**S1 File.**
(ZIP)

## Acknowledgments

We thank all the editors and reviewers in advance for their valuable comments that will improve the presentation of this paper.

## Author Contributions

**Conceptualization:** Song Qian, Liwei Yang.

**Formal analysis:** Ping Li.

**Methodology:** Yan Xue.

**Software:** Song Qian, Liwei Yang.

**Supervision:** Ping Li.

**Validation:** Song Qian.

**Visualization:** Song Qian.

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
