## [Decision Letter · Decision Letter 0]

9 Apr 2024

PONE-D-24-04252SIFusion: Lightweight Infrared and Visible Image Fusion Based on Semantic InjectionPLOS ONE

Dear Dr. Li,

Thank you for submitting your manuscript to PLOS ONE. After careful consideration, we feel that it has merit but does not fully meet PLOS ONE’s publication criteria as it currently stands. Therefore, we invite you to submit a revised version of the manuscript that addresses the points raised during the review process.

We look forward to receiving your revised manuscript.

Kind regards,

Anas Bilal, Ph.D.

Academic Editor

PLOS ONE

Journal Requirements:

"Xinjiang Uygur Autonomous Region Tianchi Talent Innovation Leading Talent Project"

Reviewers' comments:

Reviewer's Responses to Questions

**Comments to the Author**

1. Is the manuscript technically sound, and do the data support the conclusions?

Reviewer #1: Yes

Reviewer #2: Yes

2. Has the statistical analysis been performed appropriately and rigorously? 

Reviewer #1: Yes

Reviewer #2: Yes

3. Have the authors made all data underlying the findings in their manuscript fully available?

Reviewer #1: Yes

Reviewer #2: Yes

4. Is the manuscript presented in an intelligible fashion and written in standard English?

Reviewer #1: Yes

Reviewer #2: Yes

5. Review Comments to the Author

Reviewer #1: 1. Expand on how SIFusion is novel compared to existing methods, highlighting unique features and improvements.

2. Provide more detailed explanations of the Semantic Injection Module (SIM) and Edge Convolution Module (ECB), including mathematical formulations.

3. Include comparative analysis with existing image fusion methods in terms of performance metrics and computational efficiency.

4. Address the practical applicability of the method, particularly in terms of computational cost and deployment in real-world scenarios.

5. Enhance the experimental section with more diverse datasets to validate the effectiveness of SIFusion in various conditions.

6. Clarify how the method improves the processing of semantic information in high-level visual tasks, with specific examples or case studies.

7. Discuss limitations and potential areas for improvement in the method to provide a balanced view.

8. Include a section on potential future work, exploring how the method can be extended or adapted to other applications.

9. Improve the clarity and structure of the paper, ensuring that technical terms and concepts are well-explained for readers unfamiliar with the topic.

10. Strengthen the conclusion by summarizing key findings and their implications for both human and machine vision applications.

Reviewer #2: Given the limitations of existing image fusion algorithms, this paper proposes a lightweight infrared and visible light image fusion network SIFusion based on semantic injection. The method has achieved good results. Here are some suggestions:

（1）Why do the authors need to stitch together infrared and visible light images before inputting them into the model instead of extracting features separately? What is the difference between these two approaches?

（2）The author extracted fusion features and semantic features from the source image separately, but the extraction module is the same in terms of framework. Please explain how the same module extracts two different features separately.

（3）The ablation experiment is insufficient, and it is not enough to simply remove the module for comparison. It should be compared with other typical modules to demonstrate the effectiveness of the proposed module.

（4）The author claims in “Contributions” that the ECB module is their first introduction to the field of image fusion, which is incorrect. The ECB module was introduced in 2023 for infrared and visible light image fusion and was publicly published that year.

The reference is: CHEN Zhaoyu, FAN Hongbo, MA Meiyan, et al. FECFusion: Infrared and visible image fusion network based on fast edge convolution. Mathematical Biosciences and Engineering: MBE, 2023, 20(9): 16060-16082.

（5）The bold values described in Table 7 are not bolded.

（6）Because the author describes the network proposed as a lightweight network, but the paper does not compare network parameters and network computational complexity（FLOPs）, it is recommended to add this data comparison.

6. PLOS authors have the option to publish the peer review history of their article (what does this mean?). If published, this will include your full peer review and any attached files.

Reviewer #1: No

Reviewer #2: No

---

## [Author Response · Author response to Decision Letter 0]

26 Apr 2024

To Senior Editor and Reviewers，

Thank you for giving us the opportunity to submit a revised draft of the manuscript "SIFusion: Lightweight Infrared and Visible Image Fusion Based on Semantic Injection" for publication in the Journal of "PLOS ONE". We appreciate the time and effort that editors and the reviewers dedicated to providing feedback on our manuscript and are grateful for the insightful comments and valuable improvements to our paper. We have incorporated most of the suggestions made by the reviewers. Those changes are highlighted with red color in the manuscript. Please see below, for a point-by-point response to the reviewers' comments and concerns. 

Reviewer #1：

1.Expand on how SIFusion is novel compared to existing methods, highlighting unique features and improvements.

Response: Thank you for your suggestion. We have made revisions to the section on semantic guidance in the related work of this paper and have marked the changes with red color.

2.Provide more detailed explanations of the Semantic Injection Module (SIM) and Edge Convolution Module (ECB), including mathematical formulations.

Response: Thank you for your suggestion. We have added the mathematical formulas for the Semantic Injection Module (SIM) and the Edge Convolutional Block (ECB), along with corresponding explanations.

3.Include comparative analysis with existing image fusion methods in terms of performance metrics and computational efficiency.

Response: 

Thank you for your suggestion. We have added comparative analysis with existing image fusion methods in terms of performance metrics and computational efficiency. The data can be viewed in Table 5. 

It should be specifically noted that only FLFuse, TarDAL, UMF-CMGR, and the method presented in this paper are implemented based on the Pytorch architecture, while the other comparative algorithms are based on the Tensorflow 1.x architecture. The model data for these methods cannot be provided directly. Therefore, the model data is obtained through a model reconstructed based on the Pytorch architecture.

4.Address the practical applicability of the method, particularly in terms of computational cost and deployment in real-world scenarios.

Response: Thank you for your suggestion. We have made revisions to address this issue in the content of the efficiency comparison experiment.

5.Enhance the experimental section with more diverse datasets to validate the effectiveness of SIFusion in various conditions.

Response: Thank you for your suggestion. In this paper, the training set of the MSRA dataset is not directly used as the training set for the network model. Instead, data augmentation is performed by segmenting the dataset, which to some extent allows the network model to have stronger generalization capabilities. Therefore, we first conduct experimental comparisons using the LLVIP dataset. Subsequently, we also perform generalization tests on the TNO dataset and the LLVIP dataset. These datasets contain a variety of different scenarios, especially day and night scenes. Thus, we believe that the existing experiments in this paper can, to a certain extent, verify the effectiveness of SIFusion under various conditions.

6.Clarify how the method improves the processing of semantic information in high-level visual tasks, with specific examples or case studies.

Response: Thank you for your suggestion. Firstly, we illustrate the importance of semantic information for image fusion tasks and the limitations of existing methods in handling semantic information through the examples in Figure 1. Then, we introduce the design of semantic information processing in our method in the Methods section, such as the loss function for the semantic feature branch and the combination of semantic features with fusion features through SIM. Next, we demonstrate the superiority of our method by comparing it with 10 mainstream algorithms. Lastly, we validate the effectiveness of SIM through ablation experiments. In addition, we also confirm the facilitative effect of our method on advanced visual tasks through semantic segmentation experiments. Overall, we analyze the research on semantic information in our method through the above content.

7.Discuss limitations and potential areas for improvement in the method to provide a balanced view.

Response: Thank you for your suggestion. We have revised the summary section and have expanded on the content within it.

8.Include a section on potential future work, exploring how the method can be extended or adapted to other applications.

Response: Thank you for your suggestion. We have revised the summary section and have expanded on the content within it.

9.Improve the clarity and structure of the paper, ensuring that technical terms and concepts are well-explained for readers unfamiliar with the topic.

Response: Thank you for your valuable suggestion. We believe that the current manuscript has already struck a balance between professional depth and accessibility as much as possible. Specialized technical terms are all referenced with citations, allowing readers who are not familiar with the subject to gain a deeper understanding. In order to maintain the professionalism of the paper and to avoid redundancy that could arise from over-simplifying the foundational knowledge, we have decided not to make further modifications to the current version regarding this issue. If you still have difficulties understanding any of the proprietary terms, we will make further revisions to the manuscript.

10.Strengthen the conclusion by summarizing key findings and their implications for both human and machine vision applications.

Response: Thank you for your suggestion. We have revised the summary section and have expanded on the content within it.

Reviewer #2:

1.Why do the authors need to stitch together infrared and visible light images before inputting them into the model instead of extracting features separately? What is the difference between these two approaches?

Response: In the field of infrared and visible image fusion, both single-branch and dual-branch structures are among the mainstream network architectures in existing fusion methods. The dual-branch structure can extract features from two different modalities separately before fusing them, offering better interpretability. In contrast, the single-branch structure obtains shared feature information with similar information domains through the same feature extraction approach. To further achieve a lightweight network model and in conjunction with the characteristics of the semantic injection architecture, the method proposed in this paper selects a single-branch structure with fewer parameters, while the unique feature information is combined with the shared feature information through a semantic injection approach.

2.The author extracted fusion features and semantic features from the source image separately, but the extraction module is the same in terms of framework. Please explain how the same module extracts two different features separately.

Response: The two encoders used in this paper for extracting fusion features and semantic features, although they have the same network structure, do not share weight parameters. Therefore, encoders with the same structure but different parameters can learn different parameters through specific loss guidance. The encoder of the semantic branch is guided by the target loss to learn the parameters for extracting semantic features, thereby enabling these two encoders to perform different functions.

3.The ablation experiment is insufficient, and it is not enough to simply remove the module for comparison. It should be compared with other typical modules to demonstrate the effectiveness of the proposed module.

Response: Thank you for your suggestion. We have added a comparative experiment between ECB and other typical modules in the ablation experiment section.

4.The author claims in “Contributions” that the ECB module is their first introduction to the field of image fusion, which is incorrect. The ECB module was introduced in 2023 for infrared and visible light image fusion and was publicly published that year. The reference is: CHEN Zhaoyu, FAN Hongbo, MA Meiyan, et al. FECFusion: Infrared and visible image fusion network based on fast edge convolution. Mathematical Biosciences and Engineering: MBE, 2023, 20(9): 16060-16082.

Response: Thanks for pointing out the issues in our paper, and we have revised the wording of the manuscript.

5.The bold values described in Table 7 are not bolded.

Response: Thank you very much for pointing out the issue with Table 7 in our paper. We apologize for the oversight and promise to correct this error immediately. In the revised version of the paper, we will ensure that all mentioned important data and values are displayed in bold to clearly distinguish and highlight this key information.

6.Because the author describes the network proposed as a lightweight network, but the paper does not compare network parameters and network computational complexity(FLOPs), it is recommended to add this data comparison.

Response: 

Thank you for your suggestion. We have added experimental comparison data between SIFusion and ten comparative algorithms regarding runtime, operational memory space, parameter count, weight size, and FLOPs. The data can be viewed in Table 6. 

It should be specifically noted that only FLFuse, TarDAL, UMF-CMGR, and the method presented in this paper are implemented based on the Pytorch architecture, while the other comparative algorithms are based on the Tensorflow 1.x architecture. The model data for these methods cannot be provided directly. Therefore, the model data is obtained through a model reconstructed based on the Pytorch architecture.

Thank you again for the reviewers' suggestions. The revised content is marked in red within the manuscript, and the file is named "Revised Manuscript with Track Changes.pdf".

---

## [Decision Letter · Decision Letter 1]

22 May 2024

PONE-D-24-04252R1SIFusion: Lightweight Infrared and Visible Image Fusion Based on Semantic InjectionPLOS ONE

Dear Dr. Li,

Thank you for submitting your manuscript to PLOS ONE. After careful consideration, we feel that it has merit but does not fully meet PLOS ONE’s publication criteria as it currently stands. Therefore, we invite you to submit a revised version of the manuscript that addresses the points raised during the review process.

We look forward to receiving your revised manuscript.

Kind regards,

Anas Bilal, Ph.D.

Academic Editor

PLOS ONE

Journal Requirements:

Reviewers' comments:

Reviewer's Responses to Questions

**Comments to the Author**

1. If the authors have adequately addressed your comments raised in a previous round of review and you feel that this manuscript is now acceptable for publication, you may indicate that here to bypass the “Comments to the Author” section, enter your conflict of interest statement in the “Confidential to Editor” section, and submit your "Accept" recommendation.

Reviewer #2: All comments have been addressed

Reviewer #3: (No Response)

2. Is the manuscript technically sound, and do the data support the conclusions?

Reviewer #2: Yes

Reviewer #3: Partly

3. Has the statistical analysis been performed appropriately and rigorously? 

Reviewer #2: Yes

Reviewer #3: Yes

4. Have the authors made all data underlying the findings in their manuscript fully available?

Reviewer #2: Yes

Reviewer #3: Yes

5. Is the manuscript presented in an intelligible fashion and written in standard English?

Reviewer #2: Yes

Reviewer #3: Yes

6. Review Comments to the Author

Reviewer #2: (No Response)

Reviewer #3: This paper introduces SIFusion, a lightweight infrared and visible light image fusion based on semantic injection. SIFusion proposes a semantic injection module to integrate semantic information. Experimental results demonstrate its superior performance both qualitatively and quantitatively. However, I still have the following concerns about this paper:

1. In the comparison experiments, the authors compare the proposed SIFusion with other ten competitors on the MSRS and LLVIP datasets, but select six competitors on the TNO dataset. To avoid subjective bias, the authors should add experiments and propose the same comparable methods on the TNO dataset.

2. A major contribution of this work is to design a semantic injection module, thus ensuring that the fused image has rich semantic information. Fig.1 gives a schematic comparison of SeAFusion and SIFusion. Therefore, to comprehensively verify the effectiveness of the method the authors should compare some semantic-aware methods, such as SeAFusion and PSFusion, in the experiments.

3.In the fusion experiment with the MSRS dataset, the fusion performance of GANMcC better than CSF. However, it's performance of object detection is inferior to CSF. How can we explain this inconsistency between fusion performance and downstream task performance?

4. There are some errors in the references. For example, Ref.14 and Ref.37 were published in 2019 and 2022, respectively. Please double-check the whole manuscript.

7. PLOS authors have the option to publish the peer review history of their article (what does this mean?). If published, this will include your full peer review and any attached files.

Reviewer #2: No

Reviewer #3: No

---

## [Author Response · Author response to Decision Letter 1]

8 Jun 2024

To Senior Editor and Reviewers，

Thank you for giving us the opportunity to submit a revised draft of the manuscript "SIFusion: Lightweight Infrared and Visible Image Fusion Based on Semantic Injection" for publication in the Journal of "PLOS ONE". We appreciate the time and effort that editors and the reviewers dedicated to providing feedback on our manuscript and are grateful for the insightful comments and valuable improvements to our paper. We have incorporated most of the suggestions made by the reviewers. Those changes are highlighted with red color in the manuscript. Please see below, for a point-by-point response to the reviewers' comments and concerns. 

Reviewer \\#3:

This paper introduces SIFusion, a lightweight infrared and visible light image fusion based on semantic injection. SIFusion proposes a semantic injection module to integrate semantic information. Experimental results demonstrate its superior performance both qualitatively and quantitatively. However, I still have the following concerns about this paper:

1.In the comparison experiments, the authors compare the proposed SIFusion with other ten competitors on the MSRS and LLVIP datasets, but select six competitors on the TNO dataset. To avoid subjective bias, the authors should add experiments and propose the same comparable methods on the TNO dataset.

Response: Due to the TNO dataset being one of the earliest public datasets for infrared and visible light images in the military field, its content is mainly dominated by buildings, with only a small portion of images containing common semantic targets such as pedestrians and vehicles. Therefore, the proposed method did not perform well on this dataset. To better demonstrate the generalization performance of the proposed method, we replaced the TNO dataset with the newer M3FD dataset, which contains multi-scene and multi-target semantic information. We compared SIFusion with twelve comparative algorithms on the MSRS, M3FD, and LLVIP datasets, and SIFusion showed good fusion performance across all of them.

2.A major contribution of this work is to design a semantic injection module, thus ensuring that the fused image has rich semantic information. Fig.1 gives a schematic comparison of SeAFusion and SIFusion. Therefore, to comprehensively verify the effectiveness of the method the authors should compare some semantic-aware methods, such as SeAFusion and PSFusion, in the experiments.

Response: We have added comparative experiments related to SeAFusion and PSFusion, and conducted an analysis of the experiments. However, the models of SeAFusion and PSFusion require a larger number of parameters and greater consumption of computational resources. In contrast, the method proposed in this paper, while maintaining a lower computational complexity, can effectively integrate information from the source images, thereby generating fused images that are more in line with the visual effects of human vision.

3.In the fusion experiment with the MSRS dataset, the fusion performance of GANMcC better than CSF. However, it's performance of object detection is inferior to CSF. How can we explain this inconsistency between fusion performance and downstream task performance?

Response: Fusion performance metrics are used to evaluate the image quality of the fusion results and do not have a direct connection with the performance of downstream tasks. It is not possible to reflect the performance of downstream tasks through the fusion evaluation metrics. Therefore, we indirectly assess the performance of the semantic expression of the fusion results through the performance of the semantic segmentation model. From the experimental results, although the metric results of the method in this paper are not as good as PSFusion, the model parameters and computational resources required for inference of the method in this paper are far less than those of PSFusion.

4.There are some errors in the references. For example, Ref.14 and Ref.37 were published in 2019 and 2022, respectively. Please double-check the whole manuscript.

Response: Thank you for the reviewer's comments; we have re-examined the manuscript and made corrections to the relevant issues.

---

## [Decision Letter · Decision Letter 2]

28 Jun 2024

PONE-D-24-04252R2SIFusion: Lightweight Infrared and Visible Image Fusion Based on Semantic InjectionPLOS ONE

Dear Dr. Li,

Thank you for submitting your manuscript to PLOS ONE. After careful consideration, we feel that it has merit but does not fully meet PLOS ONE’s publication criteria as it currently stands. Therefore, we invite you to submit a revised version of the manuscript that addresses the points raised during the review process.

We look forward to receiving your revised manuscript.

Kind regards,

Xiaoyong Sun

Academic Editor

PLOS ONE

Journal Requirements:

Reviewers' comments:

Reviewer's Responses to Questions

**Comments to the Author**

1. If the authors have adequately addressed your comments raised in a previous round of review and you feel that this manuscript is now acceptable for publication, you may indicate that here to bypass the “Comments to the Author” section, enter your conflict of interest statement in the “Confidential to Editor” section, and submit your "Accept" recommendation.

Reviewer #2: All comments have been addressed

Reviewer #3: (No Response)

2. Is the manuscript technically sound, and do the data support the conclusions?

Reviewer #2: (No Response)

Reviewer #3: Yes

3. Has the statistical analysis been performed appropriately and rigorously? 

Reviewer #2: (No Response)

Reviewer #3: Yes

4. Have the authors made all data underlying the findings in their manuscript fully available?

Reviewer #2: (No Response)

Reviewer #3: Yes

5. Is the manuscript presented in an intelligible fashion and written in standard English?

Reviewer #2: (No Response)

Reviewer #3: Yes

6. Review Comments to the Author

Reviewer #2: (No Response)

Reviewer #3: The authors have addressed most of my concerns, but there are still some minor issues that need to be addressed.

1、The response reveals that the proposed method performed inadequately on the TNO dataset. However, concerning the MSRS, M3FD, and LLVIP datasets, the proposed SIFusion still exhibits suboptimal performance in comparison with SeAFusion and PSFusion. Hence, we suggest that the authors incorporate a Discussion section to elucidate this reason, thereby providing readers with more space for reflection.

2、There is a lack of introduction and analysis of some key works in this field.

[1] Multi-interactive feature learning and a full-time multi-modality benchmark for image fusion and segmentation

[2] Infrared and visible image fusion via interactive compensatory attention adversarial learning

[3]CDDFuse: Correlation-driven dual-branch feature decomposition for multi-modality image fusion

7. PLOS authors have the option to publish the peer review history of their article (what does this mean?). If published, this will include your full peer review and any attached files.

Reviewer #2: No

Reviewer #3: No

---

## [Author Response · Author response to Decision Letter 2]

28 Jun 2024

To Reviewers,

Thank you for giving us the opportunity to submit a revised draft of the manuscript "SIFusion: Lightweight Infrared and Visible Image Fusion Based on Semantic Injection" for publication in the Journal of "PLOS ONE". We appreciate the time and effort that editors and the reviewers dedicated to providing feedback on our manuscript and are grateful for the insightful comments and valuable improvements to our paper. We have incorporated most of the suggestions made by the reviewers. Those changes are highlighted with red color in the manuscript. Please see below, for a point-by-point response to the reviewers' comments and concerns.

Reviewer \\#3:

The authors have addressed most of my concerns, but there are still some minor issues that need to be addressed.

1. The response reveals that the proposed method performed inadequately on the TNO dataset. However, concerning the MSRS, M3FD, and LLVIP datasets, the proposed SIFusion still exhibits suboptimal performance in comparison with SeAFusion and PSFusion. Hence, we suggest that the authors incorporate a Discussion section to elucidate this reason, thereby providing readers with more space for reflection.

Response: Thank you for the reviewer's suggestions. We have added a section titled "Performance Discussion on the TNO Dataset" following the "Application of Semantic Segmentation" subsection to further discuss the shortcomings and reasons of the algorithm presented in this paper when applied to the TNO dataset.

2. There is a lack of introduction and analysis of some key works in this field.

[1] Multi-interactive feature learning and a full-time multi-modality benchmark for image fusion and segmentation.

[2] Infrared and visible image fusion via interactive compensatory attention adversarial learning.

[3]CDDFuse: Correlation-driven dual-branch feature decomposition for multi-modality image fusion.

Response: We have added introductions and analyses of these three key works in the section of related work.

---

## [Editor Report · Decision Letter 3]

2 Jul 2024

SIFusion: Lightweight Infrared and Visible Image Fusion Based on Semantic Injection

PONE-D-24-04252R3

Dear Dr. Li,

We’re pleased to inform you that your manuscript has been judged scientifically suitable for publication and will be formally accepted for publication once it meets all outstanding technical requirements.

Kind regards,

Xiaoyong Sun

Academic Editor

PLOS ONE
---

## [Editor Report · Acceptance letter]

4 Jul 2024

PONE-D-24-04252R3 

PLOS ONE

Dear Dr. Li, 

I'm pleased to inform you that your manuscript has been deemed suitable for publication in PLOS ONE. Congratulations! Your manuscript is now being handed over to our production team.

Kind regards, 

on behalf of

Dr. Xiaoyong Sun 

Academic Editor

PLOS ONE